# Hospital-Based Genomic Surveillance of *Klebsiella pneumoniae*: Trends in Resistance and Infection

**DOI:** 10.3390/biology14121795

**Published:** 2025-12-16

**Authors:** Erica Olund-Matos, Ricardo Franco-Duarte, André Santa-Cruz, Maria Nogueira, Margarida Correia-Neves, Diana Lopes, Rui Jorge Silva, Margarida Ribeiro Araújo, Inês Monteiro Araújo, Ana Filipa Martins, Carolina Maia Nogueira, Alberta Faustino, Pedro G. Cunha, Pedro Soares, Teresa Rito

**Affiliations:** 1CBMA—Centre of Molecular and Environmental Biology, Department of Biology, University of Minho, 4710-057 Braga, Portugal; 2IB-S—Institute of Science and Innovation for Bio-Sustainability, University of Minho, 4710-057 Braga, Portugal; 3ICVS—Life and Health Sciences Research Institute, School of Medicine, University of Minho, 4710-057 Braga, Portugal; 4ICVS/3B’s—PT Government Associated Laboratory, 4710-057 Braga, Portugal; 5Department of Internal Medicine, Hospital of Braga, 4710-243 Braga, Portugal; 62-CA-Braga—Clinical Academic Centre, Hospital of Braga, 4710-243 Braga, Portugal; 7Division of Infectious Diseases, Department of Medicine Solna, Karolinska Institutet, 171 77 Stockholm, Sweden; 8Clinical Pathology Service, Hospital of Braga, 4710-243 Braga, Portugal; 9Clinical Academic Centre of Hospital Senhora da Oliveira, 4835-044 Guimarães, Portugal; 10Medicine Department, Hospital Senhora da Oliveira, 4835-044 Guimarães, Portugal; 11Centre for the Research and Treatment of Arterial Hypertension and Cardiovascular Risk, Internal Medicine Department, Hospital Senhora da Oliveira, 4835-044 Guimarães, Portugal

**Keywords:** *Klebsiella pneumoniae*, multidrug resistance, virulence, epidemiology, infection, colonisation

## Abstract

*Klebsiella pneumoniae* (Kp) are bacteria that can cause serious infections, especially in hospitals, where patients are more vulnerable. These bacteria can also colonise the body without causing symptoms, but then become dangerous if the immune system weakens or other diseases emerge. Some of them have become resistant to multiple antibiotics, making treatment more difficult and increasing the risk of outbreaks. In this study, we analysed the genetic material of 115 Kp samples collected in a Portuguese hospital to understand how these bacteria spread and acquire resistance to drugs. We discovered significant diversity among the samples, including bacteria with genetic traits not previously observed in Portugal. The largest genetic group showed resistance to antibiotics usually effective against these infections, suggesting that resistance genes were transferred from other *Klebsiella* strains or even other bacteria. We also identified a bacterial variant with characteristics associated with severe disease and antibiotic resistance, which requires special attention. These results reinforce the importance of regularly monitoring bacteria circulating in hospitals and implementing preventive measures, such as careful use of antibiotics and the reduction in invasive devices. Understanding how these bacteria evolve and spread can help protect patients and prevent the aggravation of future infections.

## 1. Introduction

*Klebsiella pneumoniae* (Kp), from the *Enterobacteriaceae* family, is a globally recognised public health threat due to its rapid acquisition and spread of antimicrobial resistance and hypervirulence genes. Multidrug-resistant (MDR) Kp [1] is particularly worrying due to its ability to acquire, store and disseminate accessory genetic information through horizontal gene transfer between Kp strains and other bacterial species leading to inefficient antibiotic regimens and aggravated outcomes [2,3]. The evolutionary pressure caused by antibiotic misuse and overuse leads to new MDR and hypervirulent Kp strains [4] whose occurrence continues to rise globally [5,6], making Kp a priority pathogen for the World Health Organization (WHO), the U.S. Centers for Disease Control and Prevention (CDC) and the European Centre for Disease Prevention and Control (ECDC). Recent systematic analyses predict that antimicrobial-resistant infections, especially those caused by hypervirulent and multidrug-resistant Kp, will remain a leading cause of morbidity and mortality in healthcare facilities [1].

One particular concern is the increased frequency in reported resistance to carbapenems, broad spectrum antibiotics used to treat severe infections or *Enterobacteriaceae* with extensive resistance mechanisms, including extended-spectrum beta-lactamases (ESBLs), reducing treatment options and increasing mortality. Between 2019 and 2023, Portugal reported a 43% increase in bloodstream infections caused by carbapenemase-producing (CP) Kp [5], when compared with neighbouring countries, despite early adherence to the European Committee on Antimicrobial Susceptibility Testing (EUCAST) guidelines [1,6,7]. Fluctuating reported percentages of MDR-Kp [7] and a deficit of national and regional data in Portugal [8] undermine the understanding of trends, the implementation of hospital containment and treatment strategies, the reduction in pressure and costs to the healthcare system and the capacity of health authorities to issue evidence-based directives [9].

While genomic studies on Kp in Portugal focus on outbreaks [10], antimicrobial resistance (AMR) profiling [11], or broad national trends [12], regional studies are limited, despite their importance in understanding local resistance and virulence, as infections occur in both hospital and community settings [1]. Genomics offers insights into resistance to beta-lactams including extended-spectrum beta-lactamase (ESBL) and carbapenems [13], quinolone [8], colistin [14], sulfonamides [15], aminoglycosides [16] and tetracyclines [15], and virulence factors including capsule-regulating genes and siderophore production [17]. Strain clonality is usually assessed by Multi-Locus Sequence Typing (MLST), with gene identification through comparison to curated Kp genomic/phenotypic databases [3].

Emerging genomic studies show that hypervirulent Kp represents a rapidly evolving global threat, as epidemic lineages such as ST23, ST11, and ST258 increasingly combine resistance determinants like KPC, NDM, OXA-type carbapenemases, and colistin resistance with key virulence loci including *peg-344*, *iuc*, *iro*, and *rmpA*/*rmpA2*. International surveillance highlights frequent convergence of multidrug resistance and hypervirulence among high-risk sequence types resulting in severe outbreaks in hospitals and communities, emphasising the need for continued molecular surveillance and novel control strategies [18,19,20,21,22].

In this study, we aimed to (i) characterise the genomic diversity of Kp circulating in a major hospital in Northern Portugal; (ii) integrate whole-genome sequencing data with resistance and virulence profiles to identify high-risk lineages; and (iii) examine epidemiological and clinical patterns associated with colonisation and infection. A secondary aim was to integrate and contextualise the local genomic landscape within regional Iberian datasets. These objectives address the current lack of regional genomic surveillance data in this region of Portugal.

## 2. Materials and Methods

From December 2021 to July 2022, Kp isolates were identified using MALDI-TOF-MS (matrix-assisted laser desorption/ionisation time-of-flight mass spectrometry) in the Microbiology Laboratory at Hospital de Braga (HB), following standard European Centre for Disease Prevention and Control (ECDC) procedures. Patients over 18 were invited to participate, and clinical/epidemiological data were collected. Clinical specimens (e.g., urine, blood, wound swabs, respiratory samples) were collected by hospital personnel as part of routine diagnostic procedures and processed in the accredited microbiology laboratory according to standard institutional protocols. Only culture-confirmed Kp isolates obtained through routine clinical workflows were included in this study. Cases were classified as infection or colonisation, and infections were further classified as community-acquired or nosocomial, depending on their detection before or after the first 48 h of hospitalisation [23]. We acknowledge that this strategy may introduce a minor sampling bias as screening is not systematic in the institution and depends on voluntary patient participation.

Isolates from the same patient were included only if they corresponded to distinct tissue infections or if infections were detected in admissions over 30 days apart, in order to prioritise potentially independent sampling events. Following the General Data Protection Regulation (GDPR), all data were pseudo-anonymised. The study was approved by the HP Ethics Committee (ref. 30/2021) and by the Ethics Committee for Research in Life and Health Sciences of the University of Minho (ref. CEICVS 032/2021).

All isolates were grown from single-colony purification to avoid mixed culture artefacts. Antibiotic susceptibility testing was conducted using the VITEK 2 (bioMérieux, Marcy l’Étoile, France) and WalkAway (Beckman Coulter systems, Brea, CA, USA), following EUCAST guidelines. DNA was extracted using the Wizard^®^ Genomic DNA Purification Kit from Promega (Madison, WI, USA) (CAT. # A1120) [24]. DNA quality was assessed using Qubit^®^ and NanoDrop™ and sequenced by Novogene (Munich, Germany) on the Illumina NovaSeq 6000 to 100× coverage. Genomes were assembled with Unicycler v0.5.0 [25] that incorporates SPAdes v3.15.4 [26] and annotated using GeneMark.hmm prokaryotic v3.25 software [27] using Kp MGH 78,578 as the reference [14]. Assemblies were checked with Unicycler and SPAdes quality metrics for signs of plasmid contamination and previously described plasmids in Kp were directly searched in the raw and assembled data.

A total of 400 comparative Kp genomes from the Iberian Peninsula and the North of Africa (Portugal, Spain, and Tunisia) were downloaded (September 2023) from the Bacterial and Viral Bioinformatics Resource Center (BV-BRC) [28] using filters for quality (based on the annotation statistics and a comparison to genomes in the integrated PATRIC—Pathosystems Resource Integration Center [29]), human host, and collection time (2016–2021). Taxonomic ID quality control, and virulence and resistance profiles were conducted with Kleborate v2.2.0 [3], which assigns numerical scores for virulence (0 meaning no virulence and 5 meaning hypervirulent Kp) and resistance (0 for no acquired resistance genes detected and 3 for extensively drug-resistant profiles).

Phylogenetic trees were generated following a published pipeline [30], using a concatenated set of homologous proteins representing the estimated core genomes multi-aligned with MAFFT v7.520 [31] and Maximum Likelihood in MEGA v11 using the Dayhoff+G model with 1000 bootstraps [32]. BLASTP analysis (v.2.14.0) was performed using the full proteome of the reference genome of Kp as a query (NC_009653) against the total database, retaining homologous proteins considering an e-value cutoff of 1 × 10^−8^. Proteins present across all samples were selected for the core genome. Clusters were defined as groups of two or more genetically close HB genomes with a node distance below 0.025. This threshold was selected by visually inspecting divergence patterns within the full Iberian—North African dataset and identifying clear discontinuities between closely related groups of HB isolates and more distant background genomes. While not intended as a formal outbreak cutoff, cluster membership is interpreted conservatively and used primarily to describe local genomic relatedness.

We applied a PCAix analysis (for mixed data types) to explore the relationship between clinical variables and genomic profiles. Variables include binary (gender; use of medical devices during hospitalisation (yes/no); colonisation vs. infection, nosocomial vs. community-acquired; prior antibiotic exposure (yes/no); comorbidities (yes/no) and bacterial infections declared in the previous 12 months (yes/no), nominal (sample origin: urinary tract, central venous catheter, bone, skin/soft tissue, respiratory system, lymphatic system, abdomen, bacteraemia, other; tobacco use—active smokers, past smokers, non-smokers; alcohol intake—alcohol-dependent, former drinker, non/occasional drinker; type of residence—own residence, family residence, nursing home, foster family) and numeric (age, virulence score and resistance score, number of transfers between patient care units (PCUs) within the hospital and number of previous hospitalisations in the last 12 months). Non-parametric pairwise associations were tested using chi-square (categorical), Kruskal-Wallis (categorical vs. numeric), and Spearman’s correlation (numeric). All statistical analyses were carried out using XLSTAT (v.2024.4) [33] and R software (v.4.3.2) (www.r-project.org). Ethical approval was granted by both the Life and Health Sciences Research Institute of the University of Minho (ref. CEICVS 032/2021) and the Hospital of Braga Ethics Committee (ref. 30/2021). Local HB Kp genomes generated in this study have been deposited in the NCBI BioSample database under BioProject accession number PRJNA1232829.

## 3. Results

A total of 115 Kp genomes from 101 patients were sequenced. Most patients were elderly, with comorbidities (83.17%), undergoing nosocomial infections (86.14%) and lived in their own home (80.20%). Previous bacterial infections were reported in 34 patients, while 37 reported antibiotic intake. A total of 54.46% of the patients had no known hospitalisations, and 18.81% had at least two based on multiple reasons. Kp infections were the cause of death of 6.93% of the patients during the episode of hospital admission. Detailed information is available in Table 1.

The medical team classified Kp isolates as either colonising or infecting the patient (Table 2). Infections corresponded to 53.04% of the hospital isolates, with more than half (54.10%) being urinary tract infections. Most infections were from males (59.02%), particularly middle-aged patients, likely due to higher exposure to metabolic risk factors and greater frequency of invasive devices [34].

After quality control, a phylogeographic tree of 511 genomes was built, comprising 115 HB isolates, 395 BV-BRC genomes and reference Kp MGH 78,578 based on an estimated core genome of 2923 genes (Figure 1), in line with recent estimates [35]. The Maximum Likelihood phylogenetic tree split three different *Klebsiella* species initially identified as Kp into HB: three HB *Klebsiella Ka3* (Kka3) from the Kp Species Complex (KpSC); eight *Klebsiella variicola* subsp. *variicola* (Kvar) genomes (including four isolated HB strains) and five hundred genomes identified as Kp. Kleborate taxonomic supported the identification. Of the HB isolates, 83.48% (*n* = 96) clustered in 15 genetically close clades (11 restricted to HB) and 16.52% (*n* = 19) of HB appeared isolated. Five clusters (2, 4, 8, 9 and 12) account for more than half of the HB dataset (57.39%, *n* = 66). MLST analysis revealed 29 sequence types (STs), some associated with known high-risk lineages (e.g., ST15, ST45, ST13). A total of 12 STs occurred within the 14 Kp clusters (clusters 1 and 2 were both ST15; clusters 11 and 12 were ST45) and 17 were in the isolated genomes.

Clusters varied in virulence and resistance scores (Table 3). Seven clusters had identical virulence and resistance scores across all genomes, while the remaining showed score variation despite genomic proximity, likely due to differential acquisition of genes.

Regarding pathogenic potential, the most common O-locus types (related to immune recognition) were O1/O2v1 (in half the clusters) and O1/O2v2 (28.6%). Less frequent loci (OL103, O4, O3b) were also detected. Among isolated genomes, O1/O2v1 and O1/O2v2 were dominant (Table 3 and Appendix A). A total of 29 distinct K-loci on capsule serotype were identified, most unique to each cluster. In terms of acquired virulence, yersiniabactin was the only factor consistently present, except in one genome (cluster 4) also with colibactin.

Regarding resistance, clusters 4, 8, 12 and 14 show the highest resistance scores. Common resistance genes include aac variants (aminoglycosides), *bla*CTX-M-15 and *bla*SHV alleles (β-lactamases), and mutations in GyrA and ParC (fluoroquinolone resistance). Notably, clusters 4 (ST13), 8 (ST147), 10 (ST37), and 12 (ST45) all carry the KPC-3 carbapenemase. Although ST45, a global strain, is not typically linked to high resistance, high scores were observed in this dataset, indicating potential local adaptation. ST13 is a hypervirulent lineage usually with low resistance, but the presence of KPC-3 aligns with reports of resistant ST13 strains of likely Portuguese origin [36]. Additionally, cluster 13 (ST307) harbours both ESBL genes and a porin mutation (OmpK35), contributing to its multidrug-resistant phenotype.

Nineteen HB genomes were phylogenetically isolated. They have low virulence and genomic resistance profiles that diverged from the obtained in the HB using the VITEK 2 (bioMérieux) and WalkAway (Beckman Coulter) systems, particularly for macrolides, fosfomycin, tigecycline, and colistin (Figure 2).

Both HB antibiograms and Kleborate pointed to a standard resistance to ampicillin. Only four isolates (2, 6, 10 and 18) showed resistance to other antibiotics in the antibiograms. Three were ESBL-positive with previous exposure to relevant antibiotics. All four isolates showed carbapenem resistance in antibiograms, but only isolate 2 had corresponding genetic markers, suggesting alternative mechanisms for the resistance as efflux pump overexpression to evade drug uptake.

Kleborate detected phenicol resistance in isolates 5, 6, and 19—though these are not routinely tested phenotypically. Fluoroquinolone resistance in isolates 2 and 6 matched expected mutations in DNA gyrase and topoisomerase IV genes, while isolate 10 lacked known determinants. Isolate 6 was notably multidrug-resistant, carrying over 40% of all resistance genes found.

Several isolates (1, 7, 8, 13, 15, 16, 17) exposed previously to antibiotics did not exhibit phenotypic or genotypic resistance to specific drugs, as exposure alone does not ensure resistance. Discrepancies between genotypic and phenotypic profiles were observed, especially for macrolides, colistin, and fosfomycin (false negatives), and phenicols (false positives), underscoring the limitations of predictors due to limitations in genomic databases and gene presence not always translating into phenotypic resistance. Potential explanations for these discrepancies include variation in gene expression, the presence of resistance mutations not currently represented in reference databases, differences in plasmid copy number, or regulatory/porin-mediated resistance mechanisms known to affect carbapenems and fosfomycin in Kp [37]. Additionally, phenotypic categorisation can vary depending on EUCAST breakpoint updates. 

Across all samples, resistance genes were most frequent for tetracyclines (47.8%), fluoroquinolones (46.1%), aminoglycosides, sulphonamides, and KPC-3 carbapenemases (45.2%). *bla* genes were present in 43.48% of isolates, and ESBL-associated *bla* genes in 41.74%. Trimethoprim resistance was also seen in 41.74% of cases. Less prevalent were genes for phenicol (17.4%, *n* = 20), rifampicin (12.2%, *n* = 14), macrolide resistance (13.0%, *n* = 15), β-lactam and fluoroquinolone (both with 20%, *n* = 23), while ompK porin mutations occurred in 10.43% (*n* = 12). No acquired resistance was found for colistin, tigecycline, or ESBL inhibitors.

PCAMix analysis (Figure 3) was used to explore links between clinical, genomic, and infection/colonisation variables. All binary and numerical variables were included, while nominal variables—due to fragmentation of the variables and high levels of missing data—were reserved for pairwise comparisons. While variance explained was low (11.77%), broad trends emerged. Samples were coloured by their phylogenetic cluster (Figure 3A). We plotted the contribution of each category to the two axes (Figure 3B).

The lower-right quadrant of the PCAMix is shaped by “Number of hospitalisations”, “Previous bacterial infections”, and “Comorbidities”, with the latter two being significantly associated (*p* = 1.30 × 10^−9^). These features are not associated with specific clusters. Higher average values of hospitalisations are obtained for clusters 8 and 12. The lower-left quadrant correlates with higher resistance and virulence scores, which are themselves significantly associated (*p* = 1.76 × 10^−10^). Clusters 4 (ST15) and 12 (ST45) show the highest average scores and are concentrated here, along with members from cluster 8. These samples tend to represent nosocomial colonisations, higher resistance levels (score 2), and frequent medical device use. The most relevant variants for the upper left—right side of the plot are age, number of intra-hospital transfers, and lack of previous history of bacterial infections, with clusters 1 and 7 showing some association.

Regarding pairwise correlations, colonisation cases which had higher resistance scores (*p* = 2.76 × 10^−5^), and nosocomial infections were more often preceded by antibiotic exposure (*p* = 1.36 × 10^−4^). Older patients reported more comorbidities (*p* = 1.12 × 10^−5^). These trends are reflected along PCAMix axis F1. On the negative side, we have a trend for nosocomial, history of taken antibiotics and colonisation (clusters 1, 2, 3, 4, 8, 12) while to the right (positive values) we have the variables for community-acquired, no history of taken antibiotics and infection (most isolates, clusters 5, 6, 11, 13, Kka3). Trends can be checked in Table 3.

Further nominal variable comparisons revealed significant results for phylogenetic cluster vs. infection type (*p* = 0.00296). Isolate presence vs. sample origin (*p* < 0.00001) with colonisations mostly associated from skin/soft tissues while infections were linked to urinary and respiratory samples.

## 4. Discussion

The characterisation of 115 Kp genomes from HB, a major institution located in Northern Portugal, previously uncharacterised for Kp, enabled a comprehensive view of the local population structure and resistance patterns. Unlike studies limited to specific infection sites or resistance profiles, our approach minimised bias and allowed clear differentiation between colonisation and infection, identifying strains not actively causing disease. Integrating clinical data with genomic analysis, we uncovered links between Kp virulence, transmission routes, and infection origin (community or nosocomial). Additionally, we identified STs that had never been reported in Portugal, including some with particularly harmful combinations of resistance and/or virulence. Using WGS instead of traditional 7-loci MLST enabled more detailed observation of local evolutionary patterns, especially relevant for a species with frequent horizontal gene transfer [37]. Some concerning virulence and resistance factor combinations would be undetected without WGS. Our data reinforce recent trends in both Portugal and broader European settings, highlighting a dynamic and rapidly evolving AMR scenario. The analysis also highlights the importance of WGS for accurate species identification, since *K. variicola* was misclassified as Kp using routine diagnostic platforms. The detection of *K. variicola* among patient isolates is clinically relevant, as this species has been associated with increased mortality in some studies, despite generally lower resistance profiles compared to Kp *sensu stricto* [38].

Phylogenetic analysis revealed high genomic diversity, with most isolates forming distinct subbranches suggestive of local evolution. Some few genomes established clades with BV-BRC genomes collected from other Portuguese hospitals (within clusters 8, 11, 14), implying regional dissemination [39].

While 83.5% of HB genomes formed genetic clusters, some displayed considerable internal diversity. The low divergence observed within clusters 1 and 2 (ST15) and clusters 11 and 12 (ST45) indicates the presence of highly conserved lineages circulating in the hospital. Although such genomic homogeneity is compatible with outbreak dynamics, isolates from these clusters were recovered intermittently across the sampling period, often separated by more than 100 days, rather than appearing in temporally concentrated groups which is more consistent with long-term persistence of these clones in the hospital environment than with a short, acute outbreak. ST45 and ST15 are among the most epidemiologically significant STs in Europe and Portugal [8,11] and associated with AMR. Other reported STs include ST147 (cluster 8), ST307 (cluster 13), ST13 (cluster 4) and ST348 (cluster 14). These ST348 genomes are associated with KPC-3-carbapenemase production, a resistance pattern previously linked to an outbreak in Portugal [40]. ST lineages in cluster 5 (ST35), cluster 7 (ST20), and cluster 10 (ST37) have been previously identified in Portuguese facilities. There are also reports of resistance within ST405 (found in cluster 6) and ST29 (cluster 9) genomes isolated in Portugal [10,41], but also not epidemiologically dominant. Rare STs included ST2623 (unreported in Europe), ST1562, and ST111. ST2623 has only been reported in long-term care facilities in Northern Italy and remains an infrequent lineage without a clearly established clinical niche [42]. ST1562 has likewise been detected sporadically in clinical and surveillance collections, usually as isolated findings within larger Kp datasets, and is not regarded as a major high-risk clone [43]. ST111 has been reported at low frequency in European collections and in at least one Portuguese WGS investigation of carbapenem-resistant Kp [37,44]. ST14, a common type in Portugal [9], was notably absent, suggesting regional specificity. It also lacks the hypervirulent ST23 [5].

The presence of 16.52% of HB genomes isolated in the tree underscores the high regional genomic diversity of Kp. These isolated genomes may result from foreign introductions, from unsampled environmental sources, where screenings are rarely performed or resulting from local antimicrobial pressures. For example, isolate 6 (ST15) was genetically distant from other ST15 genomes and carried unique local resistance/virulence profiles.

Despite the established association between intestinal colonisation and subsequent infection [45], no direct progression was observed in our dataset. One patient colonised with an ST45 strain later developed a UTI (urinary tract infection) caused by a genetically distinct ST4742 strain. More than 45% of the patients had Kp colonisation, 28.70% of which were intestinal colonisation with CP-Kp, highlighting its contribution to the facility’s CP-Kp burden [46]. Nosocomial infections were predominantly UTIs, followed by abdominal and respiratory infections. Historically, Kp has a higher prevalence of respiratory infections and association of Kp with pneumonia [47]. Nevertheless, our observations align with recent studies conducted in Portuguese healthcare facilities [48]. The lower rate of respiratory infections likely reflect COVID-19 mitigation measures, including mask use, that were in place during the study [49]. The observed pattern is consistent with national data indicating that non-pharmaceutical interventions during the COVID-19 pandemic, especially mask use and physical distancing, reduced overall respiratory infections and associated hospital events in Portugal [50,51]. 

AMR profiles were highly diverse, both across and within clusters. Carbapenem resistance (45.2%) was consistent with national and European trends, while ESBL patterns aligned with other Portuguese hospitals [12,48]. The overall resistance burden observed here fits within the broader 2020–2025 pattern of rising resistance to most key antibiotics in Europe, including Portugal. Conversely, the absence of acquired colistin and tigecycline resistance in our collection contrasts with reports from other centres describing expanding colistin-resistant CR-Kp populations, and may reflect both local prescribing practices [14,52].

The presence of the KPC-3 in 47 genomes across several clusters and isolated Kvar genomes indicates ongoing CP-Kp transmission within HB. ST307, recently added to a global watchlist, did not correspond to CP-Kp in our dataset (cluster 13) but exhibited an MDR profile involving other antimicrobial families. Although ST307 in our dataset did not carry carbapenemase genes, its multidrug-resistant profile parallels European reports where ST307 has emerged as a major colistin-resistant hospital lineage, underscoring its potential to accumulate additional resistance determinants, suggesting that this lineage should be actively monitored within local surveillance programmes [14]. In contrast, ST45, not typically linked to KPC-3 (as in cluster 11), showed high KPC-3 prevalence in cluster 12. Virulent ST13 was also found with CP traits previously highlighted as worrisome [41]. Our detection of carbapenemase-producing ST13 and ST45 further supports the emergence of locally adapted high-risk clones in Portugal, overlapping with the lineages recently reported by Elias et al. and extending their distribution to Northern Portugal [36]. In opposition, some common European CP-associated STs as ST11 and ST258 were not detected [5].

Mutations in SHV genes were observed in the HB genomes, potentially undermining third-generation cephalosporin efficacy. Resistance to tetracyclines, aminoglycosides, sulfonamides, and trimethoprim was also common, likely due to selective pressure from overuse. Conversely, low resistance to phenicols and macrolides was encouraging. Notably, no acquired resistance to colistin, fosfomycin, aminoglycosides, or tigecycline was found, supporting current treatment strategies in the facility for carbapenem-resistant Enterobacteriaceae. Resistance to rifampicin, a first-line drug against regionally prevalent tuberculosis [53], was detected despite rifampicin not being used to treat Kp in this hospital, suggesting acquisition of rifampicin resistance determinants via mobile genetic elements or co-selection with other antimicrobials rather than direct therapeutic exposure in this species.

Virulence profiling showed a high prevalence of *yersiniabactin* (67%), but low frequencies of *colibactin* and *aerobactin*, both associated with hypervirulence. Capsule types (K-loci) were diverse [3]. KL62, associated with immune evasion and MDR [54], was the most frequent (36 genomes, mainly in clusters 12 and 14, where resistance was high for CP and tetracycline). Hypervirulent-associated KL57 and KL64 appeared in clusters 4 and 8, respectively—both also displaying high resistance, indicating a worrying convergence of virulence and resistance. Other K-loci lacked known associations with hypervirulence or MDR.

Although the first two PCAMix components of the epidemiological factors ac-counted for only 11.77% of the variance, the analysis helped visualise broad tendencies rather than to infer causal associations. However, we should note that these trends are concordant with global epidemiological patterns, as discussed below. The PCAMix analysis suggested that age, hospitalisation history, device use, and prior antibiotic exposure were associated with MDR Kp, particularly among colonised patients. Nosocomial isolates were more likely to have high resistance scores, while community-acquired strains often lacked resistance and virulence determinants. Previous antibiotic exposure correlated with infections caused by more resistant, nosocomial Kp lineages, supporting the role of antibiotic pressure in HB MDR selection. This aligns with the notion that hospital outbreaks are caused by specific Kp with high AMR gene prevalence [2], as in clusters 4 and 12. 

Hospitalised patients, especially older ones, are chronically exposed to Kp from the environment, surfaces, devices (catheters, ventilators), and staff, and they acquire MDR Kp passively, resulting in gut or skin colonisation rather than immediate infection. Reducing the use of devices and transfers is critical as it might mitigate the frequency of nosocomial transmissions. Our results therefore reinforce the view that controlling colonisation in high-risk wards is essential for reducing downstream infections and mortality, as highlighted in recent work on colonisation-infection dynamics in critically ill patients [14].

Community-acquired cases, in contrast, often involved younger individuals with prior hospitalisation, underlying conditions and susceptibility to bacterial infections, indicating they were not fully healthy. Such individuals are more likely to become infected by diverse environmental or community-associated strains rather than by highly resistant or virulent hospital-adapted lineages through stochastic sampling of Klebsiella diversity, consistent with their generally lower resistance and virulence determinants. This indicates that typically non-pathogenic or low-virulence strains frequently breached host defences, leading to active disease. Community-acquired Kp genomes were genetically diverse (13 STs), often appearing as isolated branches, suggesting sporadic cases lacking transmission potential.

Our epidemiological analysis results perfectly align with worldwide trends. Epidemiological risk factors known to increase susceptibility to Kp infection include both host vulnerabilities and healthcare-related exposures. Hospital-associated factors such as invasive devices, including urinary and vascular catheters, prolonged hospitalisation, intra-hospital transfers, prior antibiotic exposure and comorbidities (especially malignancy, chronic respiratory disease and immunosuppression) are well-established contributors to infection risk [46,55]. Community-level factors, including advanced age, diabetes, chronic kidney disease and alcohol use disorder, have also been associated with increased likelihood of colonisation and subsequent infection [55,56].

## 5. Conclusions

Future work should focus on long-term, longitudinal cohort studies linking colonisation and infection outcomes. Identifying high-risk colonisers at the genomic level could help prevent future infections. Adopting core genome approaches over MLST [57] will allow a higher resolution and identification of particular isolate features, supporting infection control, outbreak tracking, understanding transmission chains in the community and cost-effective treatment decisions. Although most HB genomes did not exhibit simultaneous high resistance and virulence, the risk of horizontal gene transfer across the *Klebsiella* species complex and Enterobacteriaceae family underscores the need for long-term genomic surveillance. While our results do not directly demonstrate horizontal gene transfer, the mosaic distribution of resistance determinants across unrelated lineages is consistent with the extensive plasmid-mediated gene flow that characterises Klebsiella pneumoniae evolution and the dissemination of antimicrobial resistance determinants [13].

Integrating clinical data enabled a deeper interpretation of the genomic results. The high proportion of Kp colonisations, when compared to Kp infections in the facility, suggests ongoing silent transmission and the need for tailored containment measures. Future perspectives for reducing antimicrobial resistance include the progressive integration of whole-genome sequencing (WGS) into routine clinical workflows, enabling earlier detection of high-risk lineages and hypervirulent clones, interrupting the spread at an early stage of emergence. Simultaneously, the combination of colonisation screening with genomic risk prediction can inform which patients are most likely to progress to infection, informing targeted interventions, such as device use reduction, and allowing the decrease in the emergence and dissemination of multidrug-resistant Kp in healthcare settings.

## Figures and Tables

**Figure 1 biology-14-01795-f001:**
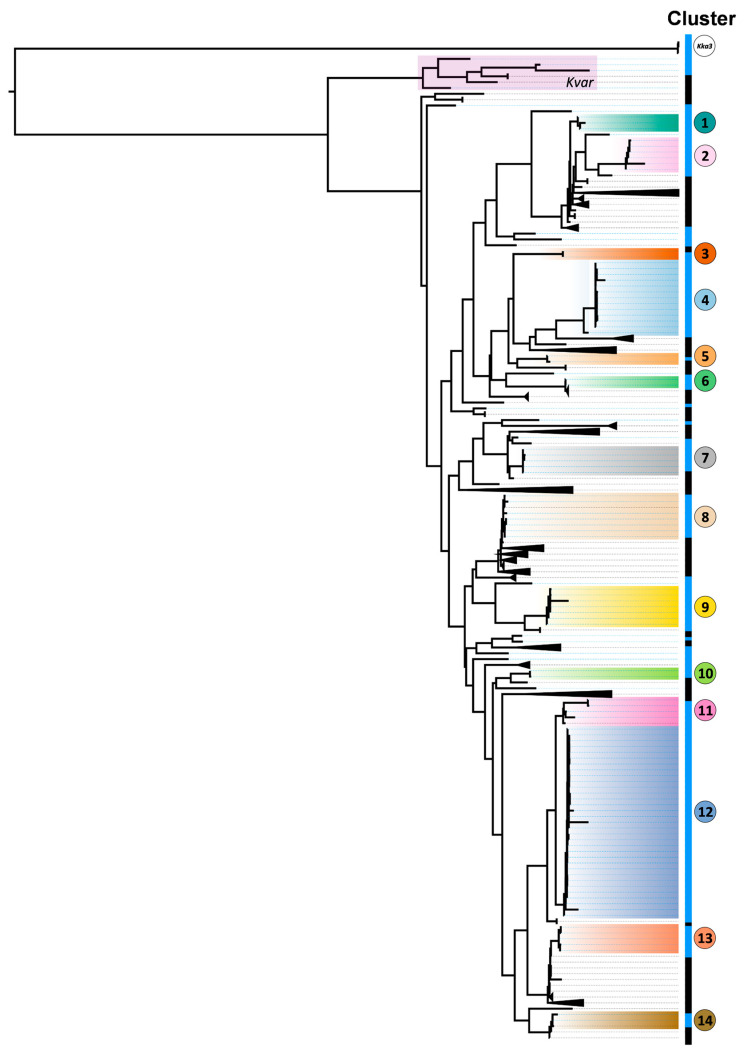
Phylogenetic tree comprising 115 local genomes (blue lateral bar) and 395 BV-BRC genomes (black lateral bar). Highlighted in different colours are 14 subclades defined by phylogenetic distance at the node less than 0.025 and labelled as clusters 1–14. The Kka3 cluster is also highlighted. Nineteen distinct blue branches with isolated Kp strains are also observed. Monophyletic clades that did not contain samples from HB were collapsed to facilitate visualisation. All collapsed clades and defined clades in the tree presented bootstrap values of 100% (Appendix A). Detailed information of each of the 14 highlighted clusters is present in Table 3.

**Figure 2 biology-14-01795-f002:**
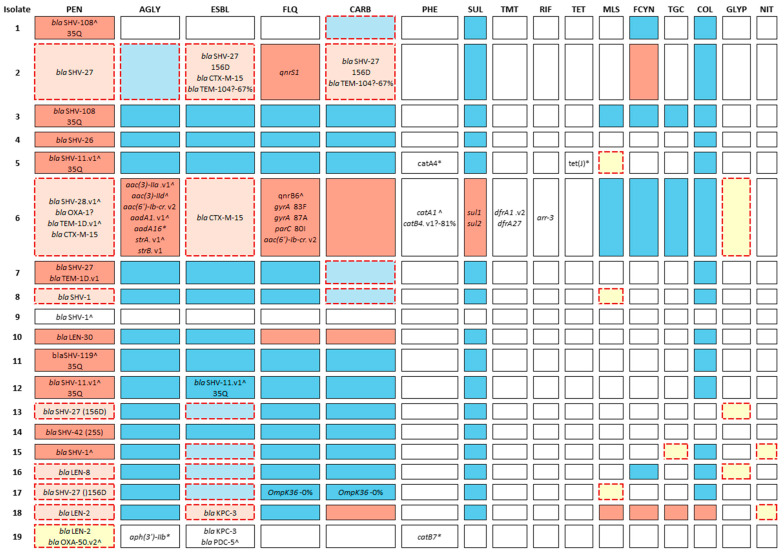
Observed resistance profiles in 19 *Klebsiella* isolates and antibiotic intake prior to isolation. Blue, red, and white indicate hospital-reported susceptibility, resistance, and missing data. Kleborate-identified resistance-associated genes are shown. A dashed square border indicates previous antibiotics contact with that drug class. Coloured cells with lighter shades (light blue/red) reflect intake from that class. Yellow corresponds to no data available, and an antibiotic for that class is being taken. Isolates 1–15 are *K. pneumoniae*, and 16–19 are *K. variicola*. Columns show antibiotic classes (PEN: penicillins; AGLY: aminoglycosides; ESBL: extended-spectrum β-lactamases; FLQ: fluoroquinolones; CARB: carbapenems; PHE: phenicols; SUL: sulphonamides; TMT: trimethoprim; RIF: rifampicin; TET: tetracyclines; MLS: macrolides; FCYN: fosfomycin; TGC: tigecycline; COL: colistin; GLYP: glycopeptides; NIT: nitroimidazole). Antibiotics taken prior to or during hospitalisation (HB) include the following: Isolate 1—meropenem (pre-HB); 2—cefuroxime, meropenem, amikacin, piperacillin-tazobactam (pre-HB); 5—azithromycin (pre-HB); 6—piperacillin-tazobactam, vancomycin, ceftazidime (in HB); 7—meropenem (in HB); 8—meropenem, amoxicillin-clavulanate, clarithromycin (pre-HB); 13—ceftriaxone, amoxicillin-clavulanate (pre-HB); piperacillin-tazobactam, vancomycin, amoxicillin-clavulanate (in HB); 15—ceftriaxone-metronidazole, ceftazidime-avibactam, tigecycline (pre- and in HB); 16—amoxicillin-clavulanate (in HB); 17—ceftriaxone-clarithromycin, piperacillin-tazobactam (in HB); 18—piperacillin-tazobactam, ceftriaxone-metronidazole (in HB); 19—piperacillin-tazobactam (in HB). Gene symbols: “?” = unconfirmed variant; “^” = low-confidence, truncated or variant hit; “*” = contains mutation potentially affecting function or resistance.

**Figure 3 biology-14-01795-f003:**
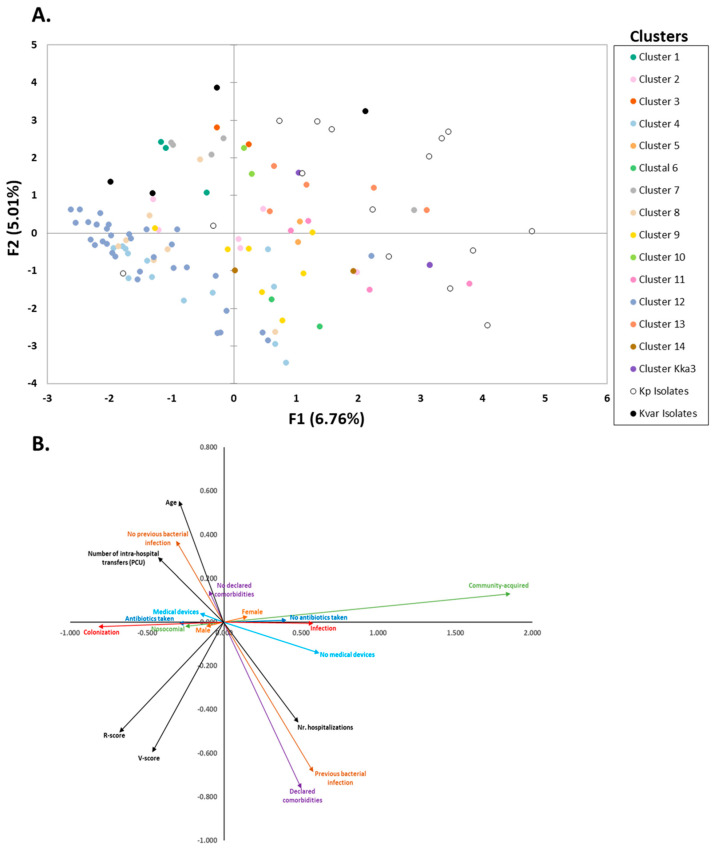
PCAMix of *Klebsiella* spp. from HB patients, integrating clinical, collection, and genomic variables: (**A**) PCAMix plot with each point representing a sample, coloured by phylogenetic cluster, previously described in Figure 1 and Table 3; (**B**) contributions of epidemiological variables to the first two PCAMix components. Binary categories (gender; use of medical devices; colonisation vs. infection, nosocomial vs. community-acquired; prior antibiotic exposure; comorbidities and previous bacterial infections) share a colour; numerical variables (age, virulence score and resistance score, number of transfers between patient care units) are in black.

**Table 1 biology-14-01795-t001:** Socio-demographic and clinical characteristics of 101 patients diagnosed with *Klebsiella* spp. infection included in this study. Categories marked with * indicate the existence of unknowns in the data.

Patient Characterisation			Patients (*n* = 101)	Female (*n* = 40)	Male (*n* = 61)
Age		Young age (18–44 years old)	8	4	4
		Middle age (45–64 years old)	28	6	22
		Old age (over 65 years old)	65	30	35
Type of *Klebsiella* sp. Infection		Nosocomial	87	34	53
		Community-acquired	14	6	8
Lifestyle	Tobacco use *	Active smokers	8	0	8
		Past smokers	15	0	15
		Non-smokers	64	29	35
	Alcohol intake *	Alcohol-dependent	10	0	10
		Former drinker	13	0	13
		Non/occasional drinker	60	28	32
	Type of residence	Own residence	81	29	52
		Family residence	16	9	7
		Nursing home	3	2	1
		Foster family	1	0	1
Comorbidities	Chronic lung diseases (e.g., COPD, asthma) *	Yes	13	4	9
		No	84	34	50
	Cardiovascular and metabolic diseases	Diabetes mellitus	39	11	28
		Hypertension	63	24	39
		Dyslipidaemia	53	22	31
	Comorbidities overall	Yes	84	32	52
		No	17	8	9
Medical history	Hospital admissions in the last year *	Yes	43	18	25
		No	57	21	36
	Hospital admissions in the last 3 months *	Yes	32	11	21
		No	68	28	40
	Antibiotic treatment in the last year *	Yes	37	15	22
		No	16	3	13
	Bacterial infections in the last year	Yes	34	14	20
		No	67	25	41
Death during hospitalisation		Yes	7	2	5
		No	94	38	56

**Table 2 biology-14-01795-t002:** *Klebsiella pneumoniae* isolates classification as either infection or colonisation and site of origin of the isolates.

Kp Isolates Classification	Origin of Isolate	Isolates (*n* = 115)	%	Female (*n* = 49)	Male (*n* = 66)
Infection	Urinary Tract	33	28.70	13	20
	Abdomen	9	7.83	3	6
	Skin and Soft Tissues	7	6.09	2	5
	Bacteraemia	5	4.35	4	1
	Bone	4	3.48	3	1
	Respiratory System	3	2.61	0	3
	*Overall*	*61*	*53.04*	*25*	*36*
Colonisation	Intestinal (CP)	33	28.70	14	19
	Urinary Tract	16	13.91	9	7
	Respiratory System	2	1.74	0	2
	Lymphatic System	1	0.87	0	1
	*Overall*	*52*	*45.22*	*23*	*29*
Unknown		2	1.74	1	1

**Table 3 biology-14-01795-t003:** Characteristics of HB genome clusters, including number of genomes (n), MLST sequence type (ST), virulence (V) and resistance (R) scores, shared virulence/resistance determinants (common resistance genes unique to a cluster are underlined), isolation sites, infection type, and classification. Asterisks indicate genes that were present in all but one (*) or two (**) genomes of the cluster. “Other” refers to sites not falling into predefined clinical categories.

Cluster (*n*)	MLST	V Score	Virulence *loci*	R Score	Resistance Determinants	Site of Isolate Collection	Nature of Infection	Isolates Classification
1 (*n* = 3)	ST 15	0	KL112, O1/O2v1, *wzi93*	1	aac(6’)-Ib-cr, qnrB1, mphA, catA1, sul1, dfrA14, OXA-1, *bla*CTX-M-15, *bla*SHV-28, *gyrA*-83F, GyrA-87A, *parC*-80I	Urinary tract|3	Nosocomial|3	Colonisation|3
2 (*n* = 6)		0 1	KL23, O1/O2v2	1	aac(3)-IIa, aac(6’)-Ib-cr, aadA16, catA1, arr-3, sul1, dfrA27, OXA-1, *bla*TEM-1D, *bla*CTX-M-15, *bla*SHV-28, *gyrA*-83F, *gyrA*-87A, *parC*-80I	Urinary tract|4 CVC|1 Not attributed|1	Nosocomial|6	Infection|4 Colonisation|1 Unknown|1
3 (*n* = 2)	ST 2623	0	KL52, OL103	0	sul2, *bla*SHV-1	Urinary tract|1 Other|1	Nosocomial|2	Colonisation|2
4 (*n* = 13)	ST 13	1 2	KL57, O1/O2v2, *ybt*+ *, *ybt*+, *clb*+	1 2	aac(3)-IIa, aac(6’)-Ib-cr, aadA2, strA, strB, qnrS1, mphA, sul1, sul2, tet(A) *, dfrA12, dfrA14 *, OXA-1, *bla*TEM-1D, *bla*CTX-M-15, *bla*KPC-3 *, *bla*SHV-1	Urinary tract|4 Bone|1 Other|8	Nosocomial|11 Community-acquired|2	Infection|2 Colonisation|11
5 (*n* = 2)	ST 35	1	KL22, O1/O2v1, *wzi37*, *ybt*+	0 1	-	Urinary tract|1 Skin/Soft tissue|1	Nosocomial|2	Infection|2
6 (*n* = 2)	ST 405	1	KL151, O4, *wzi143*, *ybt*+	1	aac(3)-IIa, aac(6’)-Ib-cr, strA, strB, qnrB1, sul2, tet(A), dfrA14, OXA-1, *bla*TEM-1D, *bla*CTX-M-15, *bla*SHV-76	Resp. system|1 Skin/Soft tissue|1	Nosocomial|2	Infection|2
7 (*n* = 5)	ST 20	0	KL39, O1/O2v1, *wzi160*	1	aac(3)-IIa, aac(6’)-Ib-cr, strA, strB, qnrB1, sul2, dfrA14, OXA-1, *bla*TEM-1D, *bla*CTX-M-15, *bla*SHV-187	Urinary tract|2 Resp. system|1 Other|2	Nosocomial|4 Community-acquired|1	Infection|3 Colonisation|2
8 (*n* = 7)	ST 147	0 1	KL64, O1/O2v1, *wzi64*, *ybt*+ *	0 2	*bla*SHV-11, 35Q, *bla*KPC-3 *, *gyrA*-83I, *parC*-80I	Urinary tract|3 Resp. system|1 Other|3	Nosocomial|7	Infection|2 Colonisation|5
9 (*n* = 7)	ST 29	1	KL19, O1/O2v2, *wzi19*, *ybt*+	1	aac(3)-IIa, aac(6’)-Ib-cr, strA, dfrA14, *bla*CTX-M-15	Urinary tract|4 Skin/Soft tissue|1 Bone|1 Abdomen|1	Nosocomial|6 Community-acquired|1	Infection|5 Colonisation|2
10 (*n* = 2)	ST 37	1	KL38, O3b, *wzi96*, ybt+	0	*bla*SHV-11, 35Q, *bla*KPC-3 *, *gyrA*-83I, *parC*-80I,	Urinary tract|2	Nosocomial|2	Infection|2
11 (*n* = 4)	ST 45	0 1	KL24, O1/O2v1, *wzi101*, *ybt*+ **	0 1	strA, strB, sul2, *bla*TEM-1D	Urinary tract|3 Resp. system|1	Nosocomial|2 Community-acquired|2	Infection|3 Colonisation|1
12 (*n* = 33)		1	KL62, O1/O2v1, *wzi149*, *ybt*+	0 2	tet(D) *, *bla*KPC-3 **	Urinary tract|10 Abdomen|1 Bacteraemia|1 Skin/Soft tissue|1 Other|16	Nosocomial|32 Community-acquired|1	Infection|11 Colonisation|21 Unknown|1
13 (*n* = 5)	ST 307	0	KL102, O1/O2v2, *wzi173*	0 1	aac(6’)-Ib-cr, aadA16, qnrB6, catII.2 *, arr-3, sul1, sul2 *, dfrA27, *bla*TEM-1D *, *bla*CTX-M-15 *, *bla*SHV-28, OmpK35, *gyrA*-83I, *parC* -80I	Urinary tract|5	Nosocomial|4 Community-acquired|1	Infection|4 Colonisation|1
14 (*n* = 2)	ST 348	1	KL62, O1/O2v1, *wzi94 **, *ybt*+	1 2	aac(3)-IIa, strA, strB, qnrB1, sul2, dfrA14, *bla*TEM-1D, *bla*CTX-M-15, *bla*SHV-11, 35Q	Urinary tract|1 Skin/Soft tissue|1	Nosocomial|1 Community-acquired|1	Infection|2
Kka3 (*n* = 3)	-	0	-	0	-	Bacteraemia|1 Bone|1 Other|1	Nosocomial|3	Infection|2 Colonisation|1

* except 1; ** except 2.

## Data Availability

The local HB Kp genomes generated in this study have been deposited in the NCBI BioSample database under BioProject accession number PRJNA1232829.

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
