# Peer review of "Hospital-Based Genomic Surveillance of Klebsiella pneumoniae: Trends in Resistance and Infection"

_biology, 2025, doi:10.3390/biology14121795_

Round 1
Reviewer 1 Report
Comments and Suggestions for Authors
This study conducted hospital based whole-genome sequencing from a Portuguese hospital to examine antimicrobial resistance, virulence, and epidemiological patterns. It identified new sequence types and demonstrated ongoing horizontal gene transfer of resistance genes. The research emphasizes local genomic surveillance to monitor the emergence of multidrug-resistant and hypervirulent strains. Its findings highlight actionable strategies for infection control and antibiotic stewardship.
Research has future application for detecting drug resistance and exploring reducing different strategies of drug resistance against important pathogens like Klebsiella pneumoniae effecting public health.
The following recommendations are proposed to further improve and strengthen the manuscript:
- Add justification of research by adding public health significance of Klebsiella pneumoniae in introduction
- How did the study verify the accuracy of horizontal gene transfer detection in light of possible plasmid contamination or mixed sequencing samples, and to what extent can the limited number of isolates reliably represent the national genomic trends of Klebsiella pneumoniae in Portugal?
- The manuscript contains minor typographical, grammatical, and punctuation errors, including incorrect hyphenation, inconsistent spelling, and missing commas. Revisions should correct fragmented sentences, maintain consistent use of “colonisation,” and improve clarity and flow throughout the text.
- “It means the bacteria often crossed host defenses, causing active disease”, should be rephrased as “This indicates that the bacteria frequently breached host defences, leading to active disease.” Similarly, “Despite the known link between intestinal colonisation and later infection [33] no direct progression was observed in our dataset” should be rewritten as “Despite the established association between intestinal colonisation and subsequent infection [33], no direct progression was observed in our dataset.”
- Add future perspective of research in reducing drug resistance in discussion section
- Which important epidemiological riosk factors enhance Klebsiella pneumoniae infection in hospitals community. Add detail in discussion section
- Add and compare outcomes of research work with recent studies (2020-2025) on drug resistance to Klebsiella pneumoniae isolates in discussion section.
Reviewer 2 Report
Comments and Suggestions for Authors
In this study, authors characterized 115 Klebsiella pneumoniae isolates from a regional hospital in Braga, Portugal between December 2021 and July 2022 using whole-genome sequencing (WGS) and clinical-epidemiological data. The authors combine genomic, phenotypic, and basic clinical data to explore the diversity, antimicrobial resistance (AMR) profiles, and potential transmission patterns of Klebsiella pneumoniae. The topic is important for regional Klebsiella surveillance however it has some concerns. Addressing these concerns will help to increase paper’s merit.
Major comments:
- In table 2 authors presented origin of isolates from various sites. However, there is no mention of how these samples were collected. Please include a section in the methods for sample collection.
- How antibiotic resistance profile was generated? Did authors performed broth microdilution or disc diffusion to confirm antibiotic resistance of the clinical isolates?
- Inclusion/exclusion criteria for repeated isolates from the same patient is not clear. Please clarify it clearly.
- What are the different subclades represented in Figure 1. Please provide what are these 14 subclades along with numbers or as a legend in the figure.
- Figures (particularly Figure 1 and 3) are not clear and could also benefit from clearer legends.
- Calling of Table 3 is way up than where Table 3 appears in the paper. Please keep Table 3 at the same page of just after mentioning it in the text for better visualization.
- In line 41, Italicize genus species name.
- Define all acronyms when they first appear. For example: AMR in line 89, MALDI-TOF-MS in line 103 and ECDC in line 104, etc.
- Fix spacing in line 132.
- Provide Cat# for all the reagents used. For example: DNA purification kit from Promega (line 115).
Reviewer 3 Report
Comments and Suggestions for Authors
The manuscript by Erica Olund-Matos and colleagues presents a genomic surveillance study of Klebsiella pneumoniae from a large Portuguese hospital. By integrating whole genome sequencing with antimicrobial resistance profiles and detailed clinical epidemiology for over one hundred isolates, it addresses a significant gap in regional data. The study is timely and methodologically sound, the manuscript has clear and logical structure and is well written. Its integration of colonization versus infection status and community versus hospital acquisition provides considerable value, as does the identification of major high risk clones and emerging carbapenem resistant lineages. These findings are important and directly relevant for local infection control and national surveillance. However, the manuscript has a number of issues that should be addressed by the authors.
Comments:
- Introduction. Lines 89-97 mention common antibiotic resistance genes and the presence of different sequence types, but do not report the results of international studies on the genomics of antibiotic resistance and multidrug resistance, which is the focus of the present manuscript. Furthermore, the Introduction does not discuss particularly aggressive sequence types and specific resistance determinants. Please describe the problem of hypervirulent Kp in an international context and connection with genomics.
- Lines 120-128. What criteria, including specific thresholds or values, was used for choosing the genomes? Which criteria for selecting Kp genomes, beyond assembly quality, were applied?
- Lines 129-130. How were the core genes collected and defined?
- Table 1 contains statistics that are interesting in their own right. Could you suggest why there is such an excess of male patients in general and in the middle age? How does this pattern correlate with the results of the genomic analysis?
- Figure 1. The tree lacks bootstrap support values. Please revise.
- Line 201. A total of 2923 genes seems too high for a core genome. Please check and explain.
- Figure 3. Please improve the resolution.
- Lines 299-300. Please check the correctness of comma placement.
- Discussion. As far as I understand, this section should show how our understanding of the subject has changed after the study. However, the beginning of the Discussion contains a long list of results and analyses without sufficient interpretative discussion and without highlighting the most important conclusions, which could be drawn, especially given the decent sample size. I suggest that the Discussion should be more clearly focused on wider comparison of your results in the context of other studies, including international work.
Overall, this is a solid genomic surveillance study that provides valuable insights into the population structure, antimicrobial resistance and virulence of Klebsiella pneumoniae in a Portuguese hospital setting, serving as a useful model for other regions. To strengthen its impact, the manuscript would benefit from clarifications on several points and a more detailed discussion of the findings in an international context.
Round 2
Reviewer 1 Report
Comments and Suggestions for Authors
I have reviewed the revised manuscript and found that Manuscript is much improved and authors have addressed all suggestions and comments and manuscript is Acceptable in Resent form. Thanks
Author Response
Many thanks for your review and comments.
Reviewer 3 Report
Comments and Suggestions for Authors
The authors made a good job in revising the manuscript and they answered all my previous comments. The changes improved the quality of the paper and make it more clear. I have only one small comment: the condition of selecting the core genome is still not fully clear. Please indicate BLAST E-value cutoff in Lines 155–157.
Author Response
Many thanks for your review and comments. I added your suggestion:
"Reviewer 3. The condition of selecting the core genome is still not fully clear. Please indicate BLAST E-value cutoff in Lines 155–157."
Thank you, we added “, retaining homologous proteins considering an e-value cutoff of 1e-8.” And we added a reference on the description of the pipeline we used.